# Novel positional encodings to enable tree-based transformers

**Vighnesh Leonardo Shiv**
Microsoft Research
Redmond, WA
vishiv@microsoft.com

**Chris Quirk**
Microsoft Research
Redmond, WA
chrisq@microsoft.com

## Abstract

Neural models optimized for tree-based problems are of great value in tasks like SQL query extraction and program synthesis. On sequence-structured data, transformers have been shown to learn relationships across arbitrary pairs of positions more reliably than recurrent models. Motivated by this property, we propose a method to extend transformers to tree-structured data, enabling sequence-to-tree, tree-to-sequence, and tree-to-tree mappings. Our approach abstracts the transformer's sinusoidal positional encodings, allowing us to instead use a novel positional encoding scheme to represent node positions within trees. We evaluated our model in tree-to-tree program translation and sequence-to-tree semantic parsing settings, achieving superior performance over both sequence-to-sequence transformers and state-of-the-art tree-based LSTMs on several datasets. In particular, our results include a 22% absolute increase in accuracy on a JavaScript to CoffeeScript translation dataset.

## 1 Introduction

### 1.1 Sequence modeling

Neural networks have been successfully applied to an increasing range of tasks, including speech recognition and machine translation. These domains crucially depend on techniques for modeling streams of audio and text, represented as dynamically sized sequences of tokens. Researchers have historically handled such data primarily with recurrent techniques, which encode sequences into fixed-length representations. The sequence-to-sequence LSTM model (Sutskever et al., 2014) is a particularly notable example in recent times.

Recurrent architectures have some disadvantages. From a generalization perspective, recurrent cells face the challenge of learning relationships between tokens many time steps apart. Attention mechanisms are now commonly employed to mitigate this issue, driving new state-of-the-art results in difficult tasks such as machine translation (Wu et al., 2016). From an efficiency standpoint, recurrence does not lend itself to parallelism, often rendering recurrent models expensive to train. Recurrent models are also difficult to interpret, employing an obtuse series of neural layers between time steps that render relationships modeled within the data unclear.

The transformer (Vaswani et al., 2017) is a stateless sequence-to-sequence architecture motivated by these issues, constructed by forgoing recurrence altogether in favor of extensive attention. This design allows information to flow over unbounded distances during training and inference, without the need for complex gates and gradient clipping. This type of long distance flow, driven by learned attention transforms over positional encoding, provides a powerful computational mechanism. Transformers also lend themselves to easier interpretation, as their attention layers can at least reveal information about learned relationships between elements of a sequence.

## 1.2 Hierarchical modeling

Recent work has begun to apply neural networks to programming tasks (Allamanis et al., 2018). In recent years, programming language analysis techniques have begun to exploit statistical techniques commonly used on large natural language corpora (Hindle et al., 2012). These can be used to identify idioms in software, enable searching for code clones, searching code by natural language, or even translating from one programming language to another.

Representing programs is an interesting challenge. One option is to view them as a one dimensional sequence of tokens and use techniques common in the natural language programming literature. However, these programs are intentionally endowed with hierarchical structure; using purely sequence-oriented methods may result in losing valuable structural information.

Expanding past sequential modeling, a common approach is to pass information through neighbors in the graph, in a manner that is reminiscent of message passing in graphical models (Li et al., 2016). To ensure that information can fully propagate across the graph, this message-passing must be applied multiple times, bounded by the diameter of the graph. While this allows us to exploit hierarchical structure, ideally we would like to do so while capturing the efficient information flow and other benefits of transformer models.

In this work, we generalize transformers to embed tree representations. Our work introduces novel positional encodings for tree-structured data[1]. Using these encodings, we can apply transformers to tree-structured domains, allowing information to percolate fully across the graph in a single layer. This can potentially extend the transformer to settings ranging from natural language parse trees to program abstract syntax trees. We evaluate our tree-transformers on programming language translation tasks such as translating JavaScript to CoffeeScript (Chen et al., 2018) as well as semantic parsing tasks including extracting a database query from a natural language request (Dahl et al., 1994), demonstrating improved performance over sequential transformers.

# 2 Positional encodings in attention models

The order of a sequence is rich in information; order-agnostic (bag-of-words) models are limited in power by their inability to use this information. One particularly common way to capture order is through recurrence; recurrent models inherently consider the order of an input sequence by processing its elements sequentially. As transformers forgo recurrence, they require information about the input sequence's order in some other form. This additional information is provided in the form of positional encodings. Each position in the input sequence is associated with a vector, which is added to the embedding of the token at that position. This allows the transformer to learn positional relationships, as well as relationships between the token embedding and positional encoding spaces.

## 2.1 Properties

The transformer's original positional encoding scheme has two key properties. First, every position has a unique positional encoding, allowing the model to attend to any given absolute position. Second, any relationship between two positions can be modeled by an affine transform between their positional encodings. The positional encodings take the form

$$
\begin{aligned}
PE_{pos,2i} &= sin(pos/f(i)) \\
PE_{pos,2i+1} &= cos(pos/f(i))
\end{aligned}
$$

where $f(i) = 10000^{2i/d_{model}}$ for $i \in [0, d_{model}/2)$. Considering the identities

$$
\begin{aligned}
cos(\alpha + \beta) &= \cos(\alpha)\cos(\beta) - \sin(\alpha)\sin(\beta) \\
sin(\alpha + \beta) &= \sin(\alpha)\cos(\beta) + \cos(\alpha)\sin(\beta)
\end{aligned}
$$

we can see that transformer can attend to relative offsets using linear transforms. For instance, the encoding of position $x + y$ can be phrased as a linear combination of $x$ and $y$'s positional encodings:

$$
\begin{aligned}
PE_{x+y,2i} &= \sin\left((x+y)/f(i)\right) = \sin\left(x/f(i) + y/f(i)\right) \\
&= \sin\left(x/f(i)\right)\cos\left(y/f(i)\right) + \cos\left(x/f(i)\right)\sin\left(y/f(i)\right) \\
&= PE_{x,2i}PE_{y,2i+1} + PE_{x,2i+1}PE_{y,2i} \\
PE_{x+y,2i+1} &= \cos\left((x+y)/f(i)\right) = \cos\left(x/f(i) + y/f(i)\right) \\
&= \cos\left(x/f(i)\right)\cos\left(y/f(i)\right) - \sin\left(x/f(i)\right)\sin\left(y/f(i)\right) \\
&= PE_{x,2i+1}PE_{y,2i+1} - PE_{x,2i}PE_{y,2i}
\end{aligned}
$$

## 2.2 Bag interpretation

Positional encodings address the power limitations of bag-of-words representations by upgrading the bag of words to a bag of annotated words. Indeed, the transformer's core attention mechanism is order-agnostic, treating keys as a bag. The calculations performed on any given element of a sequence are entirely independent of the order of the rest of that sequence in that layer; this leaves most of the work of exploiting positional information to the positional encodings (Vaswani et al., 2017), though decoder-side self-attention masking and autoregression also play a role.

Now, a bag of words annotated with positions can be equivalently thought of as a bag of positions annotated with words. From this perspective, we see that it is not at all necessary that our input "sequence" of positions have any direct correspondence with the sequence of associated "indices," i.e. an evenly distributed number line. While the original transformer's positional encodings do form this correspondence for the purposes of sequence modeling, we can consider alternative positional encodings to represent non-sequential structures in a positional space. We use this idea to extend the transformer to tree-structured data, representing structural relationships between elements as relationships between points in positional space.

# 3 Tree positional encodings

Now we construct our positional encoding scheme for trees. We focus on directed trees with ordered lists of children. Each node has a unique parent (besides the root node) and a numbered finite list of children. Each node's position can be defined as its path from the root node, and paths between nodes can climb up through parent relationships or down through child relationships.

## 3.1 Properties

In building tree positional embeddings, we aim to preserve the properties described in Section 2.1 with our new scheme. While the uniqueness property needs no adjustment, the positional relationship property needs to be modified to suit trees rather than sequences. In the context of sequences, the relationship between two positions is simply the distance $j$ that separates them. For trees though, the relation between two nodes is a *path*: a series of steps along tree branches, with each step either going up to the parent or down to a child.

Therefore, our desired property is that for all paths $\phi$, there is a corresponding affine transform in the positional space $A_\phi$ that captures the same relationship. Specifically, if $a$ and $b$ are two positions in a tree such that the path between them is $\phi$, then we desire the following:

$$
PE_b = A_\phi PE_a
$$

This allows the transformer to learn path-wise relationships within its embedding layers.

From a given node in an $n$-ary tree, there are $(n + 1)$ potential length-1 paths: a step down to any of its $n$ children, and a step up to the parent. Any longer path $\phi$ can be built as a composition of these length-1 paths.

In the positional space, we will associate the step down to child positions $1, \ldots, n$ with the affine operators $D_{1,\ldots,n}$, and the step up to the parent with the affine operator $U$. For any path $\phi$, we can construct the corresponding transform $A_\phi$ as a composition of $D$'s and $U$'s. For example, if we wish to denote the positional encoding of node $x$'s grandparent's first child (e.g., the path

Figure 1: Example computations of positional encodings for nodes in a regular tree. The sequence of branch choices $b$ determines a sequence of transforms $D_{b_1}, D_{b_2}, \ldots$ to apply to the root node's positional encoding. $U$ is defined as the complement: applying it to any node results in that node's parent (e.g. $r = Ux = U^2 y = U^3 z$). The transforms $D_i, U$ are defined in Equations 2 and 3.

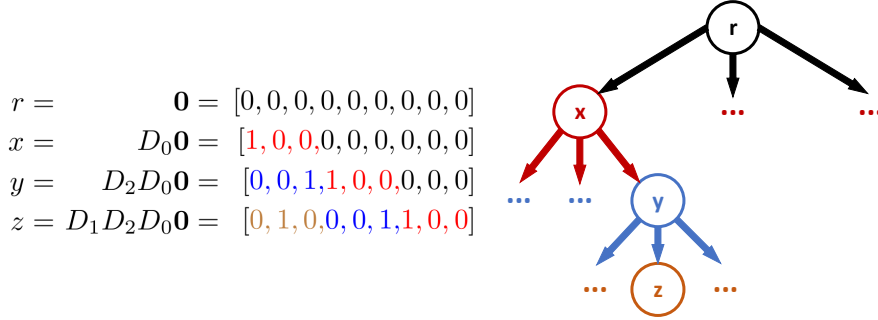

$$
\begin{aligned}
r = && \mathbf{0} &= [0,0,0,0,0,0,0,0,0] \\
x = && D_0 \mathbf{0} &= [1,0,0,0,0,0,0,0,0] \\
y = && D_2 D_0 \mathbf{0} &= [0,0,1,1,0,0,0,0,0] \\
z = && D_1 D_2 D_0 \mathbf{0} &= [0,1,0,0,0,1,1,0,0]
\end{aligned}
$$

Figure 2: Nearest neighbor heatmap of parameter-free tree encoding scheme. We number the nodes in the tree according to a breadth-first left-to-right traversal of a balanced binary tree: position 0 is the root, 1 is the first child of root, 2 is the second child of root, 3 is the first child of the first child of root, and so on. In each case, we consider the row position as a "query" and each column position as a potential "value". The attention score of solely the positional encoding after softmax is represented as a heatmap scaling from black (0.0) through red and yellow to white (1.0).

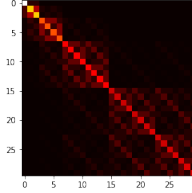

$\phi = \langle \text{PARENT}, \text{PARENT}, \text{CHILD-1} \rangle)$, we can write:

$$PE_{\text{CHILD-1}(\text{PARENT}(\text{PARENT}(x)))} = PE_{\phi(x)} = A_\phi PE_x = D_1 U^2 PE_x$$

As every path can be broken down into a composition of these $(n+1)$ operators, we need only focus on these basic operators' relationships. The fundamental relationship between these operators is that traveling up a branch negates traveling down any branch. Our constraint then is:

$$UD_i = I \; \forall i \in \{1, \ldots, n\} \tag{1}$$

### 3.2 Proposed encoding scheme

We propose a stack-like encoding for tree positions. This scheme adheres to the above constraints for all trees up to a specified depth, and still works well in practice for even deeper trees. We will start by describing a parameter-free version of our positional encoding scheme for simplicity. Our scheme takes two hyperparameters: $n$, the degree of our tree, assumed to be regular; and $k$, the maximum tree depth for which our constraint is preserved. Each positional encoding has dimension $n \cdot k$, and each operator $U, D_{1,\ldots,n}$ preserves this dimensionality. The root position is encoded as the zero vector, and every other node position is encoded according to its path from the root vector. As paths from the root consist only of steps downward, we can denote this path as $\langle b_1, \ldots, b_L \rangle$, where $b_i$ is the step choice at the $i$th layer and $L$ is the layer at which the node resides. Then, for any node $x$, we compute its positional encoding as demonstrated in Figure 1:

$$x = D_{b_L} D_{b_{L-1}} \ldots D_{b_1} \mathbf{0}$$

Now we define $D_i$ and $U$. The intuition behind our positional encoding scheme is to treat the positional encodings as a stack of length-1 component paths. Every $D_i$ operation pushes a length-1

Figure 3: Common traversals and mixtures thereof can be represented as linear transforms. Using the position encoding described in this paper, finding the parent, left child, or right child of a given node can be represented as linear transforms $U$, $D_1$, and $D_2$. Complex traversals can be represented also as linear transforms by composing these operations. The attention heatmaps below demonstrate the similarity of tree positional encodings applied to different points in the tree when the "query" has been transformed before dot product with the value.

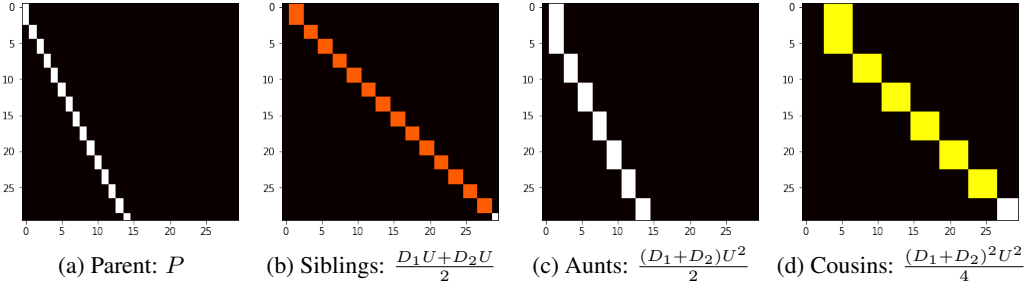

$$\text{(a) Parent: } P \qquad \text{(b) Siblings: } \frac{D_1U + D_2U}{2} \qquad \text{(c) Aunts: } \frac{(D_1+D_2)U^2}{2} \qquad \text{(d) Cousins: } \frac{(D_1+D_2)^2U^2}{4}$$

path onto the stack, while $U$ pops an element. The stack can contain at most $k$ component paths. To the extent that our assumption holds that $L \leq k$, these properties enforce Equation 1. In more explicit terms, for a given node $x$ we compute $D_i x$ by concatenating a one-hot $n$-vector with hot bit $i$ ($e_i^n$) to the left side of $x$, and truncating $x$ on the right to preserve dimensionality. We define $U$ complementarily. In other words, for a given node $x$,

$$D_i x = e_i^n ; x[: -n] \tag{2}$$
$$U x = x[n :] ; \mathbf{0}_n \tag{3}$$

where ; represents concatenation, and $[n :]$ and $[: -n]$ represent truncation by $n$ elements on the left and right side, respectively (as per Python notation). Figure 2 depicts visually how this parameter-free positional encoding scheme distinguishes between different nodes, and Figure 3 demonstrates how this scheme can efficiently represent specific structural relationships.

While these $D, U$ satisfy our constraint whenever $L \leq k$, it should be noted that for $L > k$, $UD_i$ is not necessarily the identity. Traveling down more than $k$ layers will cause this scheme to "forget" nodes more than $k$ layers up, which cannot be inverted. In practice, we make the simplifying assumption that this loss of information is insignificant for sufficiently large $k$.

The positional encoding scheme as proposed so far approximately fulfills both the uniqueness and linear composition properties. This scheme is parameter-free; however, we find that adding a parametrizable component helps diversify our encodings, improving their inductive bias. Our encoding consists of a sequence of one-hot chunks, each representing a different layer of the tree. One will note that we can weigh these one-hot chunks with any geometric series without disrupting the affine property:

$$x' = x \odot (\mathbf{1}_n ; \mathbf{p}_n ; \mathbf{p^2}_n ; \dots )$$

$x'$ here satisfies the same properties as $x$. Here, $p$ is a parameter and $\mathbf{p}_n$ is a $n$-vector of $p$'s. Figure 4 demonstrates how different values for $p$ can radically alter attention biases. Analogous to the original transformer's combination of sinusoidal encodings, we propose concatenating multiple tree encodings, each equipped with its own $p$ to be learned. To prevent the encodings' norms from exploding, we apply $\tanh$ to p to bound it between -1 and 1, and multply the encodings by a factor of $\sqrt{1 - p^2}$ to approximately normalize it. We then scale it further by a factor of $\sqrt{d_{model}/2}$ to achieve norms more similar to the original transformer's positional encoding scheme.

## 4  Decoder

To accommodate a new positional encoding scheme, we need to slightly modify the decoder. The original transformer's decoder concatenates a start token to the beginning of the sequence without modifying the positional encodings. This results in misalignment between autoregressed outputs and positional encodings, e.g. the encoding for the second position is summed with the embedding of the first output. This is not an issue in the sequential case; the positional encodings are self-similar, so

Figure 4: Nearest neighbor heatmaps of parameterized tree encodings with example values of $p$. As shown in Figure 2, many of the lower-level positions in the tree are quite similar in the absence of a decay factor. For example, position 5 (Root,$D_2$,$D_1$) is most similar to itself (score of 0.44), but quite similar to position 6 (Root,$D_2$,$D_2$) and position 3 (Root,$D_1$,$D_1$) with scores of 0.16. An appropriate level of decay allows each position to be uniquely identified as in (a); too much decay provides little additional information as in (b).

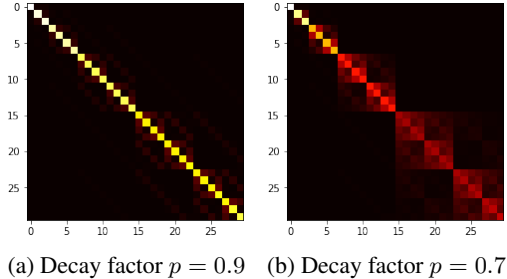

(a) Decay factor $p = 0.9$     (b) Decay factor $p = 0.7$

this "misalignment" is a linear transform away from the "correct" alignment. However, no traversal through a tree's nodes have this self-similarity property, so proper alignment here is critical.

We use a zero vector for the start token's positional encoding, and use the appropriate positional encoding for each autoregressed output. Our decoder must dynamically compute the new positional encoding whenever it produces a token. The decoder must keep track of the partial tree structure that it constructs, to correctly traverse to the next position based on history. In order to build this partial tree structure, the decoder must be aware of how many children each node must have. To this end, we construct our vocabularies such that each symbol is annotated with a number of children. When symbols have a varying number of children, they are added multiple times to the vocabulary, each with a different annotation. Given this information, the decoder can flexibly construct trees using any tree traversal algorithm, as long as it is applied consistently. In our experiments, we explored both depth-first and breadth-first traversals for decoding.

## 5 Experiments and results

For our evaluation, we consider both tree-to-tree and sequence-to-tree tasks. Both categories test our model's ability to decode tree-structured data; the sequence-to-tree task additionally tests our model's ability to translate between different positional encoding schemes. Our tree-to-tree evaulation centers around program translation tasks, while our sequence-to-tree evaluation focuses on semantic parsing.

As our model expects regular trees, we preprocess all tree data by converting trees to left-child-right-sibling representations, binarizing them [2]. This enforces $n = 2$ for our model. We use a maximum tree depth $k = 32$ for all experiments. Unless listed otherwise, we performed all of our experiments with Adam (Kingma & Ba, 2015), a batch size of 128, a dropout rate of 0.1 (Srivastava et al., 2014), and gradient clipping for norms above 10.0.

### 5.1 Tree-to-tree: program translation

For tree-to-tree evaluation, we focused on three sets of program translation tasks from the literature to test our model against. The first set of tasks is For2Lam, a synthetic translation dataset between an invented imperative and functional language. The dataset is split into two tasks: one for small programs and one for large programs. The second set of tasks involves translating between generated CoffeeScript and JavaScript code. The data is similarly spllit, here both by program length and

Figure 5: Whole program error rates for synthetic tasks with comparisons to tree2tree LSTMs. The tree-transformer demonstrates advantages over both the sequence-transformer and tree2tree LSTM, which become particularly clear for the long-sequence dataset. This indicates that the custom positional encodings may be providing useful structural information.

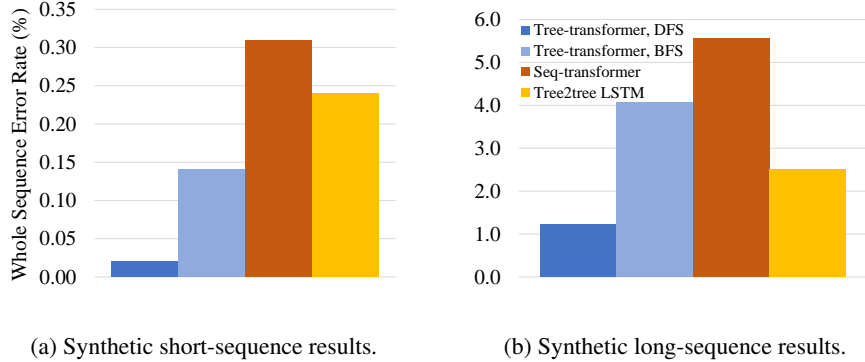

(a) Synthetic short-sequence results.  (b) Synthetic long-sequence results.

vocabulary. Each set of tasks contains 100,000 training examples and 10,000 test examples total. More details about the data sets can be found at Chen et al. (2018). We report all results in terms of whole program accuracy.

### 5.1.1  Synthetic translation tasks

For the synthetic translation tasks, we trained two tree transformers on parse tree representations, one using depth-first traversal and the other using breadth-first. We also trained a classic sequence-transformer on linearized parse trees. Both models were trained with four layers and $d_{model} = 256$. The sequence-transformer was trained with $d_{ff} = 1024$ and a positional encoding dimension that matched $d_{model}$, in line with the hyperparameters used in the original transformer. The tree-transformer, however, was given a larger positional encoding size of 2048 in exchange for a smaller $d_{ff}$ of 512. This was to emphasize the role of our tree positional encodings, which are inherently bulkier than the sequential positional encodings, while maintaining a similar parameter count.

The results for the synthetic tasks can be found in Figure 5, with comparisons to the state-of-the-art system (Chen et al., 2018). All methods compared get very close to solving the small program dataset. The results on long programs are of more interest: both tree-transformer models perform significantly better than the sequence-transformers, suggesting that the positional encodings help considerably for larger trees. The depth-first search variant outperforms breadth-first search in both cases. We conjecture that depth-first search may be a more favorable traversal method in general; it tends to construct more subtrees similar to each other earlier in the process. The depth-first variant also outperformed the tree2tree LSTM on both synthetic datasets, suggesting that the transformer's attention-based approach may be of value to program language translation just as it is to natural language translation.

### 5.1.2  CoffeeScript-JavaScript translation

Given the results on the synthetic tasks, we focused on training depth-first traversal tree-transformers for this task. The data is partitioned four ways, into two sets of vocabulary ('A' and 'B') and two categories of program length (short and long). We use the same hyperparameters as in the synthetic tasks, and once again compare our results with the tree2tree LSTM model. For memory-related reasons, a batch size of 64 was used instead for the tasks with longer program lengths.

The results for CoffeeScript-Javascript translation can be found in Figure 6. The tree-transformer obtains state-of-the-art results on over half the datasets, while still producing competitive results on the other datasets. This results demonstrate that the advantages of the tree-transformer's design are more prominent with large data. While its performance tends to be slightly weaker on the simpler short-sequence tasks, the tree-transformer gains up to 20 percentage point improvements over the

Figure 6: Whole program error rate data for CoffeeScript-JavaScript translation tasks. Here, the tree-transformer is compared to Chen et al.'s tree-to-tree model (Chen et al., 2018) which has previously produced state-of-the-art results. The tree-transformer improved results on over half the datasets, demonstrating the largest gains on the most difficult datasets.

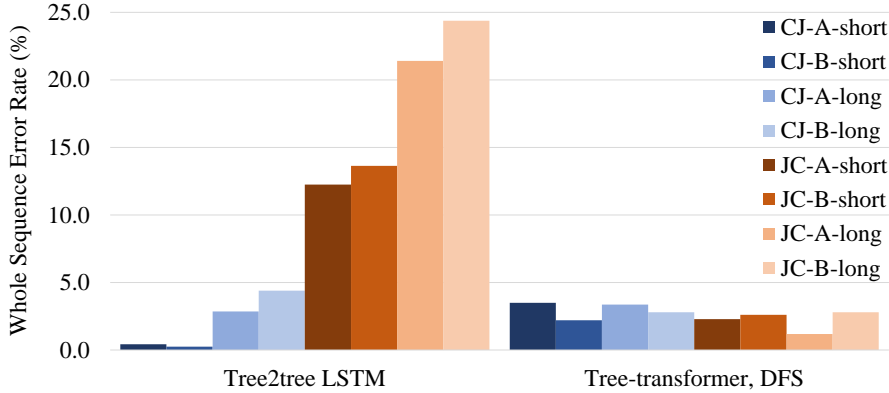

Table 1: Metrics for semantic parsing tasks. The sequence-to-tree transformer achieves state-of-the-art results on the largest dataset studied here, and outperforms the baseline transformer by several points on two of the three datasets. This suggests that the induced bias of explicit tree structure outweighs the additional hurdle of converting between positional encoding schemes. Both transformers saw less success on the two smaller datasets relative to the literature, perhaps indicating a tendency to overfit.

| Dataset | Seq2Tree Tform | Seq2Seq Tform | Literature |
|---------|----------------|---------------|------------|
| JOBS | 84.3 | 85.0 | **90.7** (Liang et al., 2011) |
| GEO | 84.6 | 81.1 | **89.0** (Kwiatkowski et al., 2013) |
| ATIS | **86.4** | 84.4 | 84.6 (Dong & Lapata, 2016) |

tree2tree LSTM on the most difficult tasks here. Overall, these results are promising for applying tree-transformers to larger-scale tree-to-tree scenarios.

## 5.2 Sequence-to-tree: semantic parsing

For sequence-to-tree evaluation, we focused on several benchmark semantic parsing tasks. In each task, the model must transform natural language queries into tree-structured code snippets against a particular query language or API. The three datasets we consider are:

- JOBS (Califf & Mooney, 1999), a job listing database retrieval task. This dataset consists of 500 training examples and 140 evaluation examples of Prolog-style query extraction.

- GEO (Tang & Mooney, 2001), a geographical database retrieval task. This dataset consists of 680 training examples and 200 evaluation examples of lamba-calculus based semantic parses.

- ATIS (Dahl et al., 1994), a flight booking task. The most informative results are for ATIS, where there is ample training data, featuring 4480 training examples and 450 evaluation examples. Each data point pairs a sentence with an argument-identified lamba-calculus expression.

JOBS and GEO provide far less data, each fewer than 1000 training examples, so their results are less reliable.

We provide whole program accuracy as our key metric to properly compare our model against the literature. For all datasets, we train four-layer sequence-to-tree and sequence-to-sequence transformers with $d_{model} = 256$, $d_{ff} = 1024$, and $d_{pos} = 2048$.

The results for our semantic parsing experiments can be found in Table 1. Here, we compare our metrics against the best in the literature as surveyed by Dong & Lapata (2016). We see that our model generally outperforms the classic transformer, leading by several percentage points on ATIS and GEO and performing only slightly worse on JOBS, the smallest evaluated dataset. It may be possible to improve our results on this smaller dataset through a cross-validated hyperparameter search, though we do not explore that here. Enforcing hierarchical structure upon the transformer appears to be worth the additional challenge of converting between positional encodings. The sequence-to-tree transformer outperforms state-of-the-art recurrent methods on ATIS, but suffers relatively on the smaller datasets. This perhaps indicates that the transformer-based approach is more prone to overfitting,

## 6    Related work

Although ours is the first effort in applying transformer models to hierarchically shaped data, there has been a range of prior work in tree-structured extensions of recurrent architectures. Soon after the recent resurgence of recurrent neural networks over linear sequences, researchers began to consider extensions of these models that accommodate structures more complex than linear chains. Initial efforts focused on input tree structures, where the shape of the input tree is fixed in advance. Tree-LSTMs demonstrated benefits in tasks such as sentence similarity, sentiment analysis (Tai et al., 2015), and information extraction (Miwa & Bansal, 2016). With a few changes, these models can be extended to cover graph-like structures as well (Peng et al., 2017).

Sequence-to-sequence models without explicit tree modeling have been applied to tree generation using only a simple linearization of the tree structure (Vinyals et al., 2015; Eriguchi et al., 2017; Aharoni & Goldberg, 2017). Later work has proposed generation methods that are more sensitive to tree structures and well-formedness constraints (Dong & Lapata, 2016; Alvarez-Melis & Jaakkola, 2017), leading to new-state-of-the-art results.

Rather than explicitly modeling hierarchically structured data, some recent work imposes hyperbolic geometry on the activations of neural networks (Gulcehre et al., 2018). Defining attention in terms of hyperbolic operations allows modeling of latent hierarchical structures. In contrast, our work focuses on the case of explicit hierarchical structure. Another recent method imposes implicit hierarchical structure over linear strings using ordered neurons. (Shen et al., 2019). The ordering can be interpreted as a fuzzy hierarchy over recurrent memory cells, with commonly-forgotten neurons corresponding to deeper tree nodes and long-retained neurons corresponding to nodes closer to the root. A hierarchy over words can be reconstructed given that activations of the model's master forget gate. This method is appropriate for imposing an inductive bias, but is less suited for scenarios that require strict enforcement of tree structures.

A particularly relevant transformer variant explicitly captures relative position, rather than relying on sinusoidal models to indirectly model distances (Shaw et al., 2018). However, this clear precursor to modeling labeled, directed graphs is limited to relative linear positions.

## 7    Conclusion

We have proposed a novel scheme of custom positional encodings to extend transformers to tree-domain tasks. By leveraging the strengths of the transformer, we have achieved an efficiently parallelizable model that can consider relationships between arbitrary pairs of tree nodes in a single step. Our experiments have shown that our model can often outperform sequence-transformers in tree-oriented tasks. We intend to experiment with employing the model on other tree-domain tasks of interest as future work.

By abstracting the transformer's positional encodings, we have established the potential for generalized transformers to consider other nonlinear structures, given proper implementations. As future work, we are interested in exploring alternative implementations for other domains, in particular graph-structured data as motivated by structured knowledge tasks.

Finally, in this paper we have only considered binary trees: in particular, binary tree representations of trees not originally structured as such. Arbitrary tree representations have their own advantages and complications; we would like to explore training on them directly.

## Footnotes

[1]Implemented in Microsoft ICECAPS: `https://github.com/microsoft/icecaps`

[2] Although binarizing trees may not always be necessary, both programming language and natural language trees of ten have constructs with unbounded numbers of children (e.g. statement blocks). For years, natural language parsing efforts have converted n-ary grammars into binary forms to enable efficient algorithms and estimation (Klein & Manning, 2003). We explored left-child-right-sibling representations primarily to be consistent with past work (Chen et al., 2018); it would be interesting to measure the impact of alternate binarization strategies (or omitting binarization altogether) when using tree transformers.

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
