[Reviews · NeurIPS 2019]

Reviewer 1



I think it is an interesting paper that proposes novel positional encodings for Transformer model applied on tasks that use trees as inputs/outputs. The authors evaluated their model on variety of tree based tasks and convincingly demonstrate improvements over baselines. I recommend its acceptance

Reviewer 2



Originality: To my knowledge, the proposed tree positional encodings are novel. They are fairly straightforward, using concatenated one-hot embeddings for each level of the tree (below the maximum depth). Quality: Some terms are inappropriate. In particular, the transformations between embeddings at different layers are not linear or affine (for which there is a unique inverse). The experiments are sufficient and demonstrate the viability of the approach. The authors obtain state-of-the-art results on some, but not all tasks considered. The authors correctly mention that the composition property breaks down past the maximum depth. Uniqueness also fails at this point, but there are no experiments evaluating whether this can be a significant issue in practice. Clarity: I had some issues understanding the paper. The term "linear" or "affine" was particularly confusing. In figure 1, it would be helpful to show the actual representations, in addition to the sequence of transformations. Datasets are often introduced with little details, referring to other papers. Significance: As sequential data isn't always the most appropriate representation, extending the transformer models to other data structures could be useful for many tasks.

Reviewer 3



ORIGINALITY : (good) This paper presents a novel positional embedding for transformer networks. The definition of this embedding, specifically designed to support tree-like datasets, is a novel contribution. Interesting perspective at the end of Section 2 with the bag-of-word and bag-of-position perspective. It is clear how this work differs from previous work. Related work is adequately cited, but some sections of the paper were missing credibility by the lack of citations, in particular, line 72-74 and 87-89. In general, any claim about key properties such as independence or other mathematical concepts should be supported by appropriate references. QUALITY : (average) Overall this submission is of good quality, but non-negligible improvements need to be considered. Claims are not all supported by theoretical analysis or experimental results. For instance, I was not convinced about the following claims: - line 72-74: "The calculations performed on any given element of a sequence are entirely independent of the order of the rest of that sequence in that layer; ...". - line 88: "any relationship between two positions can be modeled by an affine transform ...". - line 107,108: "... and k, the maximum tree depth for which our constraint is preserved." - line 117-119: It is not clear at all why this is the case All such claims need to be better supported by some evidence (either theoretical or experimental or at least by citing previous work). While introducing a novel mechanism, this submission could be more convincing with more experiments, in particular with more complex tree structures. This was properly acknowledged by the authors. Some sections of the paper need better motivation; in particular the paragraph between lines 121 and 126. the "lack of richness" is not enough motivation to add complexity to the already complex encoding. Did the parameter-free positional encoding scheme didn't work at all? In any case, experimental evidence is required to motivate this choice. Eventually, the abstract mentions the proposition of a method "enabling sequence-to-tree, tree-to-sequence and tree-to-tree mappings." however, this submission didn't cover anything on tree-to-sequence mapping. CLARITY : (low) - The submission is technically challenging to understand. Section 3 needs to be revised to make it easy to follow. For instance, between line 111 and 112, x is defined as a function of D_i's; but a few lines below, in equation (1) D_i is defined as a function of x. A figure (in addition to Figure1) or other notations could help make this section clearer. - The abstract mentions the proposition of a method "enabling sequence-to-tree, tree-to-sequence and tree-to-tree mappings." However, this submission didn't cover anything on tree-to-sequence mapping. - Figures 1, 2 and 4 are not mentioned in the text. If the authors want the readers to look at them and read through their long captions, they should be mentioned in the text. - Most (if not all) citations should be reformated. In general, all citations should be completely in parenthesis like so: "this and that is true (author, year).". Note the year is not again in ( ). Inline citations should be used when the authors of the cited paper are part of the subject of a sentence; for instance: "author (year) did this and that.". In that case, only the year is in ( ) . - some typos: "multply" (l.134) ; "evaulation" (l.158) SIGNIFICANCE (average +): The results provided in this work are positive and suggest that the proposed method is powerful. However, additional experiments and a more detailed analysis could help confirm this. Other researchers could find this work interesting as it is the first to propose a helpful inductive bias for transformers to handle tree-like datasets. The main limitation for adoption would be the clarity of the work. Figures and schema could help. The authors address a difficult task in a better way and hence improve the state of the art for two translation tasks (one being synthetic), and for one semantic parsing task.

[Author Response · NeurIPS 2019]

Thank you to the reviewers for their helpful comments.

**Reviewer #3**

Regarding affine relationships between positions: we agree that a transformer's layers do not produce simple affine
transformations. However, each layer's query and key transforms are affine. In the original transformer, the sinusoidal
positional encodings can index a relative position in the sequence using a linear transformation: say position $pos$
is represented by a positional encoding $PE_{pos}$, then positional encoding of $pos + k$ can be reconstructed using a
transformation matrix $A_{+k}$, so $PE_{pos+k} = A_{+k}PE_{pos}$, allowing query and key transformations to model relative
postitions. (This is explained in more detail in our response to Reviewer #4.) This lets the model easily attend to relative
positions along the sequence.

In our paper, we want our tree positional encoding scheme to share this property, so that the model can easily attend to
relative positions in the tree. Each individual move up or down the tree can be captured with an affine transformation,
and a series of moves can be represented as the composition of their affine transformations. We will expand Figure 1 to
include example positional encodings and the matrices that transform them, generally review Section 3 carefully.

Overall, we'll make a careful pass of this section of the paper to improve readability. We'll also provide further
background about each of the datasets considered.

**Reviewer #4**

Regarding lines 72-74, which claim that the calculations of any element of a sequence are independent of the order
presented, we can cite the original transformer paper (*Attention is all you need*), which says this in section 3.5: "Since
our model contains no recurrence and no convolution, in order for the model to make use of the order of the sequence,
we must inject some information about the relative or absolute position of the tokens in the sequence. To this end, we
add "positional encodings" to the input embeddings at the bottoms of the encoder and decoder stacks." The XLNet
paper (https://arxiv.org/pdf/1906.08237.pdf) also exploits this property: holes can be introduced in the middle of the
sequence, because order is captured using only positional encodings.

Regarding line 88, which claims that relationships between sequential positional encodings are modeled by affine
transformations, we can again cite the original transformer paper: "We chose this function because we hypothesized
it would allow the model to easily learn to attend by relative positions, since for any fixed offset $k$, $PE_{pos+k}$ can be
represented as a linear function of $PE_{pos}$." The original paper defines positional encodings as follows:

$$PE_{(pos,2i)} = \sin\left(pos/10000^{2i \ / \ d_{model}}\right); PE_{(pos,2i+1)} = \cos\left(pos/10000^{2i \ / \ d_{model}}\right)$$

Relative movements in position encodings can leverage the following identities:

$$cos(\alpha + \beta) = \cos(\alpha)\cos(\beta) - \sin(\alpha)\sin(\beta); sin(\alpha + \beta) = \sin(\alpha)\cos(\beta) + \cos(\alpha)\sin(\beta)$$

The transformation that attends to a position offset by $k$ is a block diagonal matrix, using the identities above to shift
each position accordingly.

Regarding lines 107-108 and 117-119 (the tree transformations and their preservation of movements up to depth $k$)
we'll spell this out in more detail. In effect, our positional encoding acts like a stack, where a downward movement
pushes on a particular child position to the end of the stack, and an upward movement pops off the last position from
the stack. Because our stack has fixed dimension, once we push too many elements onto the stack, we lose information
about the root.

Regarding "lack of richness": we did find that parameter-free approach did not work as well in practice. Richness is
perhaps a poor term here, as the parameters don't add extra computational power over the initial projection matrix.
Rather, the parameters induce a bias during training time by correlating nodes at the same depth. We believe this bias
helps the model incorporate the kind of information seen in the example heatmaps included in our figures. We will
rewrite this section to explain this intuition better in the revised draft.

We'll improve the notation around lines 111 and 112 to clarify that the $D_i$ and $U$ operations can act upon any node,
and we will expand Figure 1 to include an example of positional encodings that result from applying these operations.
Please note that in Equation 1, $D_i$ is not defined as a function of x – rather, the left-hand side is the matrix-vector
product of $D_i$ and $x$.

Although we did not evaluate the positional encodings in any tree-to-sequence setting, one can use tree positional
encodings on the encoder side and the original sequential positional encodings on the decoder side to represemt
tree-to-sequence mappings.

Thank you for your suggestions on citing uncited figures and reformatting citations. We will incorporate these into the
revised draft.

[Meta-Review · NeurIPS 2019]

Reviewers are all on the side of acceptance, the AC recommends accepting this work as more attention on positional encodings should be of interest to the community.